# Natriuretic Peptide Levels and Stages of Left Ventricular Dysfunction in Heart Failure with Preserved Ejection Fraction

**DOI:** 10.3390/biomedicines11030867

**Published:** 2023-03-13

**Authors:** Elisa Dal Canto, Marielle Scheffer, Kirsten Kortekaas, Annet Driessen-Waaijer, Walter J. Paulus, Loek van Heerebeek

**Affiliations:** 1Laboratory of Experimental Cardiology, Division Heart & Lungs, Utrecht University Medical Centre, 3584 CX Utrecht, The Netherlands; e.dalcanto@umcutrecht.nl; 2Department of Cardiology, OLVG, 1091 AC Amsterdam, The Netherlands; m.scheffer@olvg.nl; 3Department of Cardiology, Leiden University Medical Center, 2333 ZA Leiden, The Netherlands; 4Department of Radiology, OLVG, 1091 AC Amsterdam, The Netherlands; 5Amsterdam University Medical Centers, 1066 CX Amsterdam, The Netherlands

**Keywords:** heart failure, heart failure with preserved ejection fraction, natriuretic peptides, echocardiography, cardiac magnetic resonance, fibrosis

## Abstract

In heart failure with preserved ejection fraction (HFpEF), natriuretic peptide (NP) levels are frequently lower. In several trials, the outcome differed between patients with low and high NP levels. This suggests that NP could be used to identify distinct stages of left ventricular (LV) remodeling and myocardial tissue composition. This study investigated cardiac remodeling/dysfunction and myocardial tissue characteristics assessed by echocardiography and cardiac magnetic resonance (CMR) in HFpEF patients in relation to NP levels. Clinical and echocardiographic data of 152 HFpEF patients were derived from outpatient visits. A total of 71 HFpEF patients underwent CMR-derived T1-mapping. Multivariable regression analyses were performed to examine the association of NT-proBNP categories (</> median) and NT-proBNP as continuous variable with echocardiography and CMR-derived T1-mapping. Mean age was 71 ± 9, 93% of patients were women and median NT-proBNP was 195 pg/mL, with 35% of patients below the diagnostic cut-off value (<125 pg/mL). Patients with high NT-proBNP had comparable LV systolic function and LV relaxation but significantly worse LV stiffness and left atrial function compared with patients with low NT-proBNP. Higher NT-proBNP was significantly associated with higher LV stiffness and extracellular volume fraction (ECV) (β = 1.82, 95% CI: 0.19;3.44, *p* = 0.029). Higher NT-proBNP levels identify HFpEF patients with worse LV stiffness because of more severe myocardial extracellular matrix remodeling, representing an advanced stage of HFpEF.

## 1. Introduction

Heart failure (HF) with preserved ejection fraction (EF; HFpEF) is characterized by a rising prevalence and similarly dismal outcome as HF with reduced EF (HFrEF) [1,2]. HFpEF represents a complex and heterogeneous clinical syndrome [3], and current recommendations advocate improved phenotyping of HFpEF patients to allow targeted therapies [4]. Recently, a distinct metabolic/inflammatory “obesity” HFpEF phenotype, especially prevalent in women, was identified [5,6], which corroborates the paradigm proposing comorbidities (such as obesity, diabetes mellitus, hypertension, chronic kidney disease, older age and postmenopausal state) to induce myocardial extracellular matrix (ECM) remodeling and augmented cardiomyocyte stiffness through coronary microvascular endothelial inflammation [7]. Cardiomyocyte stiffness in HFpEF is increased by the downregulation of cyclic guanosine monophosphate (cGMP)–protein kinase G (PKG) signaling due to the impaired upstream bioavailability of nitric oxide (NO) and natriuretic peptides (NPs) [8]. NP levels are frequently low in HFpEF patients, with a substantial proportion (20–30%) of patients with invasively confirmed HFpEF presenting with low to even normal values [9,10], complicating the diagnosis of HFpEF. Low plasma levels of N-terminal pro-brain natriuretic peptide (NT-proBNP) in HFpEF were attributed to metabolic comorbidities inducing a relative state of NP deficiency and decreased tissue responsiveness [5,6,11], low left ventricular (LV) diastolic wall stress due to concentric LV remodeling [12], a cushioning effect of epicardial fat, dampening LV diastolic distension [5] and postmenopausal estrogen deficiency [13]. Despite suboptimal diagnostic accuracy, NP levels carry important prognostic value and predicted adverse outcome and prognosis in HFpEF clinical trials [14,15,16]. In addition, NP levels were also associated with cardiac remodeling in HFpEF and community-based populations [17,18,19] and with therapeutic efficacy in several [14,15,20], but not all [16,21,22], HFpEF trials. In particular, NT-proBNP plasma levels correlated with the collagen volume fraction in endomyocardial biopsies from HFpEF patients [17] and with the myocardial extracellular volume (ECV) assessed by cardiac magnetic resonance (CMR) in individuals from the Multiethnic Atherosclerosis Study [18] and HFpEF patients [19]. Overall, these findings suggest that NP levels could correspond to distinct HFpEF stages, characterized by different myocardial structural and functional changes as well as clinical characteristics and prognosis. However, how measures of cardiac remodeling and dysfunction and of myocardial tissue composition relate to NP plasma levels in HFpEF patients remains incompletely understood. Accordingly, this study aims to investigate the association between clinical and echocardiographic characteristics and CMR measurements of myocardial ECV with NT-proBNP levels in HFpEF patients.

## 2. Results

### 2.1. Clinical and Echocardiographic Characteristics

The mean age of HFpEF patients was 71.2 ± 8.8 years, and 92.7% were women. There was a high prevalence of arterial hypertension (71.7%), type 2diabetes mellitus (T2DM, 34.2%) and obesity (53.9%) and frequent use of cardiovascular drugs. Thirty-four HFpEF patients had atrial fibrillation (AF, chronic (*n* = 12) or paroxysmal (n = 22), together 22.4%. Median plasma NT-proBNP value was 194.9 pg/mL (interquartile range 84.7–436.4) (Table 1). The mean H2FPEF score was 4.9 ± 1.8, and the mean HFA-PEFF score was 4.6 ± 1.1.

All HFpEF patients had evidence of LV diastolic dysfunction consisting of a prolonged DT (206.5 ± 41.5 ms), reduced E′ velocity (mean E′ = 6.5 ± 1.4 cm/s), elevated E/E′ (mean E/E′ = 13.9 ± 4.7), reduced mean A’ velocity (8.7 ± 2.1 cm/s), dilated left atrial (LA) volumes (max LA volum index (LAVI) = 43.6 ± 11.8, pre-A LAVI = 32.4 ± 9.8 and min LAVI = 25.0 ± 9.4 mL/m^2^), depressed LA reservoir function (LA global emptying fraction (ef) = 44.6 ± 9.0%) as well as LA conduit (LA passive ef = 25.2 ± 7.7%) and pump (LA active ef = 26.5 ± 9.4%) functions (Table 2).

### 2.2. Clinical Characteristics, Cardiac Structure and Function in HFpEF Patients with Low and High NT-proBNP Levels

The distribution of HFpEF patients in accordance with plasma NT-proBNP is shown in Figure 1. According to the median NT-proBNP value of 194.9 pg/mL, 34.9% of HFpEF patients fell below the diagnostic cut-off value proposed in the 2021 ESC-HFA guidelines (≤125.0 pg/mL) [23] and 70.4% of HFpEF patients fell below the NT-proBNP cut-off value recommended for risk enrichment in HFpEF trials (<360.0 pg/mL) [24], whereas 61.2% of HFpEF patients fell below the cut-off values frequently used in clinical trials (300 pg/mL).

To detect differences in clinical and cardiac characteristics in relation to plasma NT-proBNP levels, HFpEF patients were categorized into low or high NT-proBNP groups based on their NT-proBNP being lower or higher than the median value of 194.9 pg/mL (Table 1 and Table 2). HFpEF patients with high NT-proBNP were older and had worse renal function, whereas the comorbidities burden (arterial hypertension and T2DM) was similar compared with HFpEF patients with low NT-proBNP. Additionally, HFpEF patients with high NT-proBNP had a more frequent use of diuretics and beta blockers compared to those with low NT-proBNP (Table 1).

In terms of cardiac phenotype, HFpEF patients with high NT-proBNP had a similar LV geometry, LV systolic function and LV relaxation (lateral, septal and mean E′) compared with HFpEF patients with low NT-proBNP (Table 2).

However, patients with high NT-proBNP showed a significantly deteriorated LV diastolic stiffness, as evident from higher E/E′ ratios (septal and mean) at comparable LV end diastolic volume (LVEDV), additional LA enlargement (max, pre-A and and min LAVI) and additional worsening of LA function (reduced A’, LA active ef and LA compliance) (Table 2). These differences remained significant in the fully adjusted model, except for LA active ef (Table 2). When NT-proBNP was analyzed as a continuous variable, increasing values of log-transformed NT-proBNP were not significantly associated with any indices of LV remodeling, systolic function and LV relaxation in the fully adjusted analysis (Table 3).

Conversely, with increasing values of log-transformed NT-proBNP, all indices of LV diastolic stiffness deteriorated (E/E′ ratios, A’ mean, LA global and active ef and LA compliance). Additionally, increasing NT-proBNP levels were significantly associated with larger right atrial volume (RAV) (Table 3). Sensitivity analysis, which compared echocardiographic measures of patients with NT-proBNP below and above the cut-off value suggested for risk enrichment in HFpEF trials, showed differences between the groups in terms of further impaired LV stiffness, LA remodeling and dysfunction. In fact, patients with NT-proBNP > 360.0 pg/mL showed a significantly decreased A’, increased lateral E/E′, larger LA volumes and decreased LA active ef and LA compliance compared with those with NT-proBNP < 360.0 pg/mL (Appendix A). Finally, patients with NT-proBNP > 360.0 pg/mL had significantly larger RAV and showed signs of depressed right ventricular (RV) function, with reduced RV strain, and additionally reduced LV global longitudinal strain (GLS), although these differences were no longer significant in the fully adjusted analysis (Appendix A).

### 2.3. Association between Myocardial Tissue Characteristics and NT-proBNP Levels

The association between NT-proBNP as a continuous variable and CMR indices of myocardial fibrosis was investigated in 71 HFpEF patients. The clinical and echocardiographic characteristics of this subgroup reflected those of the whole study population (Appendix A). Log-transformed NT-proBNP was significantly associated with log-transformed CMR-derived extracellular volume (Table 4). The linear relationship between log-transformed NT-proBNP and ECV is represented in Figure 2. Higher NT-proBNP plasma levels were significantly and independently associated with higher proportion of ECV, β = 1.74 (95% CI:0.10–3.48).

## 3. Discussion

The present study investigated the association between cardiac remodeling and dysfunction measures and NT-proBNP plasma levels in a chronic, stable HFpEF population that consisted predominantly of postmenopausal women with a high prevalence of metabolic comorbidities and hypertension. Compared with patients with lower NT-proBNP levels, those with higher NT-proBNP levels had comparable evidence of LV concentric remodeling, systolic function and slow LV relaxation, but substantially higher LV stiffness (Table 2). The analysis of NT-proBNP as a continuous variable confirmed these results as only the associations between NT-proBNP and LV stiffness related parameters, and not those with measures of LV remodeling, systolic function and relaxation, were significant. Furthermore, we demonstrated that increased LA volumes and LA dysfunction are strongly associated with rising NT-proBNP levels, which is in agreement with LA dilatation and dysfunction being reflective of the progression of diastolic LV dysfunction [25,26] and increased filling pressures [27] in HFpEF patients.

In a subgroup of HFpEF patients undergoing CMR, increased levels of NT-proBNP were associated with more advanced myocardial ECM remodeling, evident from higher ECV. Previously, in HFpEF patients, NP levels were shown to correlate with higher LV filling pressure [28], collagen volume fraction in endomyocardial biopsies [17] and myocardial ECM remodeling quantified by CMR [19]. Furthermore, ECV as measured by CMR was independently associated with invasively measured LV stiffness moduli in HFpEF patients [29]. According to the amount of ECV, different HFpEF pathomechanisms can be assumed, with predominant myocardial stiffness in patients with increased ECV and predominant impairment of LV relaxation kinetics for those with normal ECV [29]. Taken together, our results are in line with previous findings, with different levels of NP indicating distinct stages of LV remodeling and dysfunction and myocardial tissue composition in HFpEF.

### 3.1. Relatively Low NT-proBNP Plasma Levels in “Comorbidity-Driven” HFpEF Patients

In the present study, median NT-proBNP was relatively low (194.9 pg/mL) and 34.9% of the population fell below the diagnostic cut-off value proposed by current guidelines (≤125.0 pg/mL), whereas 70.4% of the patients had a NT-proBNP level below 360 pg/mL, the cut-off level suggested for risk enrichment in HFpEF trials [24]. Conditions highly prevalent in patients with HFpEF, such as concentric LV remodeling, cardiometabolic comorbidities and postmenopausal estrogen deficiency, are associated with lower NT-proBNP levels [5,6,11,12,13]. The relatively low NT-proBNP levels in our study population consisting primarily of comorbidity-driven HFpEF patients, may thus be explained by: (1) study design criteria with recruitment of a chronic, stable outpatient HFpEF population and exclusion of cardiomyopathies; (2) recruitment of a subgroup of HFpEF patients who may be in an earlier stage of myocardial disease progression because elevated NT-proBNP was not compulsory for HFpEF diagnosis; and (3) the predominance of elderly, postmenopausal women with high prevalence of cardiometabolic risk factors.

### 3.2. NP Levels Mirror Stage of Myocardial Disease Progression in HFpEF

Despite relatively low to even normal plasma NT-proBNP levels in a subset of our HFpEF patient population, HFpEF patients with NT-proBNP levels in the lower range showed concentric LV remodeling, diastolic LV dysfunction and LA dilatation and dysfunction (Table 2). Our results are supported by a recent study, which showed that patients with invasively proven HFpEF and normal NP levels had more LV hypertrophy, worse diastolic LV function, worse LA function and a 2.7-fold higher risk for mortality or HF readmissions compared with controls [30,31]. In terms of cardiac structural and functional remodeling as well as prognosis, HFpEF patients with normal NP levels were situated between controls and HFpEF patients with high NP levels, and it was suggested that HFpEF with normal NP levels reflects an earlier stage of myocardial disease progression [30,31]. In HFpEF, myocardial stiffness is mainly determined by both the ECM and the cardiomyocytes [32,33,34]. Increased cardiomyocyte stiffness results from post-translational modifications of the giant elastic sarcomeric protein titin due to impaired upstream NP- and NO-mediated activation of cGMP-PKG signaling [8,35,36], which is proposedly inflicted by comorbidity-induced systemic inflammation and coronary microvascular endothelial dysfunction. In addition, estrogen hormone is crucially involved in cardiomyocyte lusitropic signaling as it enhances cardiomyocyte relaxation [37] and stimulates endothelial NO synthase activity to improve NO-mediated titin-based cardiomyocyte compliance [38]. On the other hand, estrogen deficiency, present in postmenopausal women, reduces NP and NO bioavailability, which compromises cardiomyocyte relaxation and distensibility [13]. Hence, high diastolic LV stiffness in HFpEF patients with lower range NP levels is more likely to result from increased cardiomyocyte stiffness rather than prominent myocardial interstitial fibrosis, and improving cardiomyocyte stiffness could therefore represent a therapeutic target. Interestingly, sodium glucose cotransporter 2 inhibitors (SGLT2i) were recently shown to improve clinical outcomes in HFpEF patients [21,22] and to ameliorate cardiac remodeling and dysfunction [39,40,41]. In human HFpEF cardiac biopsies and animal HFpEF models, SGLT2i were shown to inhibit myocardial fibrosis, hypertrophy, inflammation and oxidative stress and to improve mitochondrial function, coronary microvascular endothelial function and NO bioavailability and enhance cGMP-PKG mediated cardiomyocyte distensibility [39,40,41]. These mechanisms of action help to explain the beneficial effects of SGLT2i for HFpEF patients in general, but also suggest that SGLT2i might be especially beneficial in patients with the metabolic/inflammatory HFpEF phenotype regardless of NP levels, because of the close matching of therapeutic myocardial targets to prevailing underlying pathophysiological mechanisms.

### 3.3. NP Levels as Selection Criteria for HFpEF Trials

Markedly elevated NT-proBNP levels are used as an inclusion criterion in phase III HFpEF trials [24]. A relatively high NP cut-off level improves the diagnostic specificity and increases cardiovascular event rates in the studied population. However, setting high NP entry criteria will skew the recruited HFpEF study population towards a phenotype with more advanced ECM remodeling, in whom NP levels are significantly elevated, and will exclude a significant proportion of HFpEF patients with less advanced ECM remodeling, in whom NP levels may be lower to normal, but who still have genuine diastolic LV dysfunction, probably due to increased titin-based cardiomyocyte stiffness. The cut-off criteria for NT-proBNP levels were ≥300 pg/mL in both the EMPEROR-preserved and DELIVER trials, whereas actual median NT-proBNP levels were close to 1000 pg/mL in both trials [21,22]. According to our results, such NT-proBNP selection cut-off criteria would lead to the exclusion of 70% of HFpEF patients, who may have also been responsive to SGLT2i therapy, despite lower NT-proBNP levels. This hypothesis is currently being tested in the phase II randomized Stratified Treatment to Ameliorate DIAstolic left ventricular stiffness in early Heart Failure with preserved Ejection Fraction (STADIA-HFpEF) trial (ClinicalTrials.gov identifier NCT04475042) [42]. The STADIA-HFpEF trial recruits a more homogeneous metabolic/inflammatory phenotype HFpEF-like population with or without elevated NT-proBNP levels. HFpEF patients who have CMR evidence of structural cardiomyopathies or myocardial ECV > 29% are excluded, thereby directing enrollment towards HFpEF patients without prominent myocardial interstitial fibrosis and therefore high diastolic LV stiffness related to increased titin-based cardiomyocyte stiffness [42].

In the present study, we demonstrate that NP levels can be used to identify distinct stages of cardiac structural and functional remodeling in patients with HFpEF. Low NP levels identify an earlier stage of myocardial disease progression, with diastolic LV dysfunction characterized by impaired relaxation, LA remodeling and dysfunction and minor ECM remodeling. In contrast, high NP levels identify a more advanced stage of myocardial disease progression, with similar concentric remodeling and impaired relaxation, but elevated LV stiffness and more advanced LA and ECM remodeling. Therefore, NP levels in HFpEF could aid in improving phenotypic and pathophysiologic stratification and may potentially also be of interest for improving patient selection for individualized therapeutic inroads.

## 4. Methods

### 4.1. Study Population

Data of chronic HFpEF patients (n = 152) were derived from their routine outpatient clinic visits at OLVG hospital, Amsterdam, from January 2016 onwards. All patients presented with symptoms of dyspnoea (New York Heart Association (NYHA) Class II to III), LVEF ≥ 50% and echocardiographic evidence of LV diastolic dysfunction, according to the American Society of Echocardiography/European Association of Cardiovascular Imaging (ASE/EACVI) criteria and European Society of Cardiology—Heart Failure Association (ESC-HFA) consensus recommendation for HFpEF diagnosis [4,43]. To enhance the validation of HFpEF diagnosis, at least one of the HFA-PEFF [4] and H2FPEF [44] diagnostic probability scores had to be positive. In case of an intermediate HFA-PEFF and/or H2FPEF score, diastolic stress testing was performed using rest/exercise (cycle ergometry) right heart catheterization to confirm or reject the diagnosis. Plasma NT-proBNP was obtained at the time of echocardiography and determined with a standard immunoassay (Cobas, Elecsys NT-proBNP II, Roche, Basel, Switzerland). To minimize the confounding effects of an acute HF episode on NT-proBNP levels, the study population consisted exclusively of stable, chronic HFpEF patients without obstructive coronary artery disease (CAD). Exclusion criteria were infiltrative or hypertrophic obstructive cardiomyopathy, hemodynamically significant uncorrected obstructive or regurgitant valvular heart disease and presence of CAD evident from inducible ischemia on noninvasive testing or from a history of previous myocardial infarction. Finally, to exclude alternative causes of dyspnea, only patients with hemoglobin levels > 7 mmol/L and spirometry (FEV1/FVC > 80%) were included. The study was conducted in accordance with the Declaration of Helsinki, approved by the institutional review board of OLVG hospital, Amsterdam and data inclusion for use in research was approved by all study participants.

### 4.2. Echocardiography

Echocardiographic examinations were performed on a GE Vivid9 ultrasound machine (General Electric Medical Systems, Horten, Norway) using a specific protocol involving 2-dimensional (2D), M-mode, Doppler, tissue Doppler and 2D speckle tracking (STE) imaging in accordance with current recommendations [45]. LA volumes were assessed using the biplane area–length method from apical 2- and 4-chamber views and were indexed to body surface area (LA volume and LA volume index, LAVI). LA volumes were measured at LV end-systole (max LAVI), at the onset of A-wave mitral inflow (pre-A LAVI) and at LV end-diastole (min LAVI). LA phasic functions were calculated as: LA global ef = [(max LAVI − min LAVI)/max LAVI × 100] (reservoir function); LA passive ef = [(max LAVI − pre-A LAVI)/max LAVI × 100] (conduit function); and LA active ef = [(pre-A LAVI − min LAVI)/pre-A LAVI × 100] (pump function) [25]. LA compliance was calculated as LA global ef/E/E′ [26]. LAVI pre-A, LA passive and active ef could not be reliably assessed in 12 patients with chronic atrial fibrillation (AF), who were then excluded from the analysis for these parameters. RV systolic function was assessed from tricuspid annular plane systolic excursion (TAPSE); fractional area change (FAC), measured using the apical four chamber view on 2D echocardiography; and RV free wall strain, assessed by STE. Doppler assessment of the tricuspid valve systolic jet velocity (TR velocity) was obtained in both parasternal and apical 4-chamber views. RAV was measured at end-systole (max RAV) in apical 4-chamber view.

### 4.3. Cardiac Magnetic Resonance

CMR examination of 71 HFpEF patients was performed within 6 months from echocardiography on a Philips 1.5-T scanner (Ingenia 1.5T, Koninklijke Philips N.V., Amsterdam, the Netherlands). All scans were performed following a protocol consisting of functional analysis, T1-weighted images and late gadolinium enhancement. T1-mapping images were obtained using a modified look-locker inversion recovery (MOLLI) sequence before and fifteen minutes after intravenous gadolinium administration. Images were obtained in short-axis on basal, midventricular and apical section. Post-processing evaluation of T1-mapping values and ECV was performed using dedicated software (IntelliSpace Portal version 10, Koninklijke Philips N.V., Amsterdam, the Netherlands). T1-mapping analysis was performed on short-axis images on basal, mid, and apical slices. Images were manually contoured in native and post-contrast images by an investigator and verified by a magnetic resonance imaging level 3 cardio-radiologist. Papillary muscles were excluded from myocardial tissue. The equation used for ECV measurement was the following [46]:$$ECV=(1-hematocrit)\times \frac{{T1\mathrm{myocardium}\;\mathrm{post}\text{-}\mathrm{contrast}}^{-1}-{T1\mathrm{myocardium}\;\mathrm{native}}^{-1}}{{T1\mathrm{blood}\;\mathrm{post}\text{-}\mathrm{contrast}}^{-1}-{T1blood\;native}^{-1}}$$

### 4.4. Statistical Analysis

Data are shown as mean and standard deviation (continuous data) or as count and percentage (categorical data). Because NT-proBNP, lateral E/E′ and T1-mapping indices distributions were skewed, they are reported as medians and interquartile ranges. We categorized patients into two groups according to the median value of NT-proBNP, and we used multivariate linear regression analysis to compare the groups in terms of clinical characteristics and echocardiographic measures. For this analysis, we used predefined models to adjust for potential confounders. A minimally adjusted model included age (years), sex, plasma creatinine (mg/dL) and body mass index (BMI, kg/m^2^) (model 1). The fully adjusted model additionally included systolic blood pressure (mmHg), use of beta blockers (yes/no) and use of loop and thiazide diuretics (yes/no).

We also assessed associations of log-transformed NT-proBNP as a continuous variable with echocardiographic measures and with log-transformed CMR T1-mapping indices. A sensitivity analysis was performed by stratifying and comparing patients according to the NT-proBNP cut-off recommended for risk enrichment in HFpEF trials (=360 pg/mL) [24]. All analyses were performed using SPSS Statistics, version 22.0 (IBM Corp, IBM SPSS Statistics for Windows, Armonk, NY, USA). All reported *p* values were two-sided and values ≤0.05 were considered statistically significant.

## 5. Study Limitations

Some limitations of this study should be considered. First, sexes were not equally distributed in our study population, which mostly consisted of women. Although it is well known that women outnumber men in HFpEF, the proportion of women in our study is higher than other reports, and this might induce difficulties with the comparison of results across studies. A possible explanation for this distribution may be the applied stratification of patients, with the exclusion of patients with obstructive CAD and cardiomyopathies. Second, the size of the overall study population and of the subgroup undergoing CMR is relatively small; however, these patients were well-phenotyped with NP, a comprehensive echocardiographic analysis and myocardial tissue characteristics. Furthermore, the subgroup of patients undergoing CMR showed clinical and echocardiographic characteristics comparable to the overall study population. Third, CMR was not performed on the same day as echocardiography and biomarker assessment, which may create differences in loading conditions. However, the NT-proBNP measurement was repeated the day of the CMR examination, and this value was used for comparison with T1-mapping indices. Furthermore, our study included multiple echocardiographic markers, such as LA volumes, that reflect long-term exposure to LV filling pressure as well as measures of myocardial remodeling and tissue composition that are not affected by changes in volume status and do not vary in the short-term.

## 6. Conclusions

Compared to HFpEF patients with lower NT-proBNP plasma levels, those with higher NT-proBNP levels had comparable LV structure and LV relaxation but higher LV stiffness and ECV on CMR imaging in a study population with a comorbidity-driven HFpEF phenotype. The latter suggests that patients with high NT-proBNP have a more advanced stage of LV and LA remodeling with more severe myocardial ECM remodeling.

As NT-proBNP reflects the severity of cardiac dysfunction and remodeling in HFpEF, it may potentially serve a contributory role in improving pathophysiologic and therapeutic stratification in patients with HFpEF. 

## Figures and Tables

**Figure 1 biomedicines-11-00867-f001:**
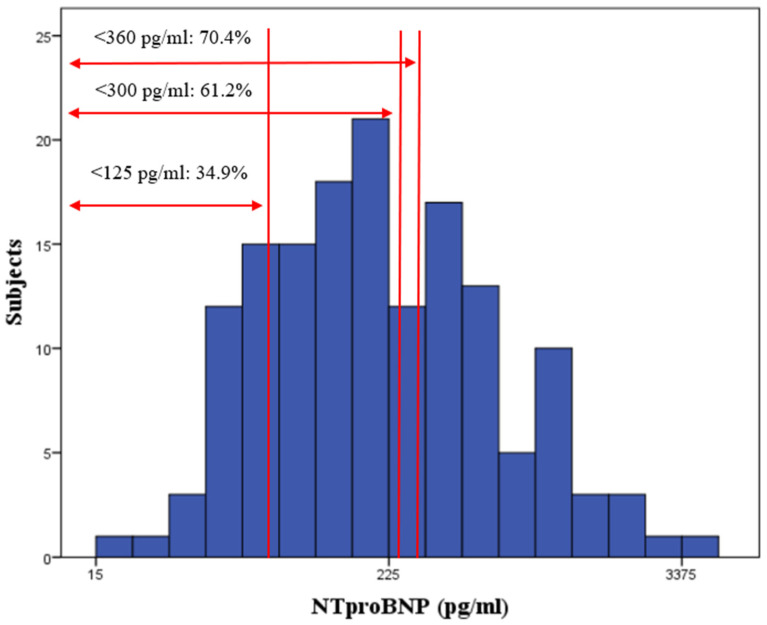
Distribution of HFpEF subjects in accordance with NT-proBNP. Median value of NTproBNP for HFpEF subjects was 194.9 pg/mL. A total of 34.9% (45 subjects) had a NTproBNP value below 125 pg/mL; 61.2% (79 subjects) had a NTproBNP value below 300 pg/mL; and 70.4% (107 subjects) had a NTproBNP value below 360 pg/mL.

**Figure 2 biomedicines-11-00867-f002:**
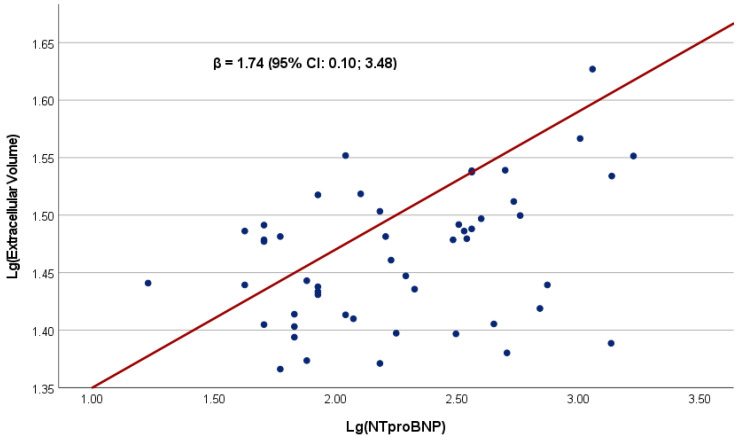
Correlation between log-transformed NT-proBNP and log-transformed Extracellular Volume in 71 HFpEF patients. The non-adjusted regression coefficient is presented.

**Table 1 biomedicines-11-00867-t001:** Clinical characteristics of the overall study population and of patients with low and high NT-proBNP (< and > the median, 194.9 pg/mL).

Parameters	HFpEF Patients (n = 152)	Low NT-proBNP (n = 78)	High NT-proBNP (n = 74)	* *p*-Value	^†^ *p*-Value
**Demographics**					
Age (yr)	71.2 ± 8.8	69.2 ± 8.5	73.0 ± 8.6	0.008	-
Sex women (%)	141 (92.7)	72 (92.3)	70 (94.6)	0.570	-
BMI (kg/m^2^)	31.7 ± 6.4	32.0 ± 5.6	31.2 ± 7.1	0.484	-
**Hemodynamics**					
Systolic BP (mmHg)	147 ± 22	142 ± 18	151 ± 25	0.008	0.032
Diastolic BP (mmHg)	77 ± 12	76 ± 10	77 ± 14	0.658	0.178
Pulse Pressure (mmHg)	69 ± 19	66 ± 17	72 ± 20	0.089	0.737
**Medical history n (%)**					
Hypertension	109 (71.7)	56 (71.7)	54 (73.0)	0.973	0.478
T2DM	52 (34.2)	24 (30.8)	28 (37.8)	0.389	0.970
Obesity	82 (53.9)	47 (60.3)	36 (48.6)	0.151	-
Atrial Fibrillation	34 (22.4)	11 (14.1)	23 (31.1)	0.014	0.070
Coronary Artery Disease	15 (9.9)	7 (9.0)	8 (10.8)	0.724	0.581
**Laboratory values**					
eGFR-EPI (mL/min/1.73 m^2^)	65.8 ± 18.8	72.7 ± 15.6	59.8.1 ± 20.0	<0.001	-
Haemoglobin (mmol/l)	8.1 ± 0.9	8.3 ± 0.7	8.0 ± 1.0	0.013	0.039
NT-proBNP (pg/mL)	194.9 (84.7–436.4)	84.7 (57.2–144.1)	423.7 (305.1–940.7)	-	-
**Medications n (%)**					
ACE inhibitors or ARBs	91 (60.0)	45 (57.7)	48 (64.9)	0.476	0.292
Loop diuretics	55 (36.2)	15 (19.2)	41 (55.4)	<0.001	<0.001
Thiazide diuretics	36 (23.7)	22 (28.2)	14 (18.9)	0.164	0.085
Aldosteron antagonists	20 (13.2)	11 (14.1)	9 4 (12.2)	0.762	0.790
Calcium channels-blockers	58 (38.2)	24 (30.8)	35 (47.3)	0.049	0.096
Beta-blockers	87 (57.2)	37 (47.4)	51 (68.9)	0.012	0.021
Oral antiglycaemic agents	41 (27.0)	18 (23.1)	23 (31.1)	0.285	0.571
Insulin	21 (13.8)	10 (12.8)	11 (14.9)	0.763	0.819
Statins	98 (64.5)	55 (70.5)	45 (60.8)	0.133	0.142

Data are shown as n (%) or mean ± SD or median (interquartile range). BMI: body mass index. BP: blood pressure. T2DM: Type 2 Diabetes Mellitus. eGFR: estimated glomerular filtration rate. NT-proBNP: N-terminal pro-brain natriuretic peptide. ACE inhibitors: angiotensin-converting enzyme inhibitors. ARBs: angiotensin II receptor blockers. The *p*-value refers to the comparison between low and high NT-proBNP groups. * Crude comparison. ^†^ Comparison adjusted for age, gender, BMI and creatinine.

**Table 2 biomedicines-11-00867-t002:** Echocardiographic measures of the overall study population and of patients with low and high NT-proBNP (< and > the median, 194.9 pg/mL).

Parameters	HFpEF Patients (n = 152)	Low NT-proBNP (n = 78)	High NT-proBNP (n = 74)	* *p*-Value	^†^ *p*-Value
**LV structure and geometry**					
LVMI (g/m^2^)	87.0 ± 17.6	84.9 ± 15.5	89.7 ± 19.5	0.096	0.268
PWTd (mm)	10.0 ± 1.5	9.9 ± 1.6	10.0 ± 1.4	0.715	0.673
RWT	0.42 ± 0.07	0.42 ± 0.07	0.43 ± 0.07	0.567	0.381
LVEDV (mL)	80.5 ± 20.2	83.1 ± 18.7	79.2 ± 28.4	0.795	0.970
LVEDVI (mL/m^2^)	42.0 ± 9.6	42.4 ± 8.7	41.8 ± 10.6	0.679	0.661
**LV systolic function**					
EF (%)	56.2 ± 5.6	56.5 ± 5.2	56.1 ± 6.3	0.659	0.632
GLS (%)	18.6 ± 2.9	18.7 ± 2.8	18.6 ± 3.1	0.856	0.465
**LV diastolic function**					
DT (ms)	206.5 ± 41.5	208.0 ± 39	204.0 ± 44	0.525	0.371
Lateral E′ (cm/s)	7.3 ± 1.9	7.4 ± 1.6	7.3 ± 2.2	0.584	0.590
Septal E′ (cm/s)	5.7 ± 1.2	5.8 ± 1.1	5.6 ± 1.3	0.245	0.941
Mean E′ (cm/s)	6.5 ± 1.4	6.6 ± 1.2	6.4 ± 1.5	0.287	0.715
Mean A’ (cm/s)	8.7 ± 2.1	9.3 ± 1.9	8.0 ± 2.0	0.001	0.030
Lateral E/E′	12.5 (9.1–14.1)	11.1 (8.4–12.8)	12.1 (10.4–15.5)	0.007	0.139
Septal E/E′	15.9 ± 5.2	14.3 ± 4.0	17.4 ± 5.9	0.001	0.107
Mean E/E′	13.9 ± 4.7	12.6 ± 3.7	15.1 ± 5.3	0.001	0.117
**LA structure and function ^$^**					
Max LAVI (mL/m^2^)	43.6 ± 11.8	39.8 ± 9.0	47.9 ± 13.1	<0.001	0.002
Pre-A LAVI (mL/m^2^)	32.4 ± 9.8	29.5 ± 7.3	35.8 ± 11.1	<0.001	0.001
Min LAVI (mL/m^2^)	25.0 ± 9.4	21.3 ± 6.3	28.7 ± 10.7	<0.001	<0.001
LA global ef (%)	44.6 ± 9.0	47.2 ± 8.4	42.3 ± 9.4	0.001	0.063
LA passive ef (%)	25.2 ± 7.7	25.2 ± 7.3	25.3 ± 8.0	0.939	0.255
LA active ef (%)	26.5 ± 9.4	29.0 ± 9.5	24.3 ± 9.6	0.005	0.153
LA compliance	3.2 ± 1.2	3.6 ± 1.2	2.8 ± 1.1	0.001	0.003
**RV and RA structure and function**					
TAPSE (mm)	22.1 ± 3.5	22.6 ± 3.2	21.7 ± 3.7	0.117	0.713
RV FAC (%)	41.9 ± 10.8	42.4 ± 12.1	41.3 ± 9.0	0.605	0.484
RV strain (%)	22.8 ± 7.1	23.7 ± 6.5	21.5 ± 7.9	0.306	0.841
TR velocity (m/s)	2.65 ± 0.40	2.60 ± 0.47	2.69 ± 0.33	0.330	0.741
Max RAV (mL)	43.2 ± 15.1	40.0 ± 11.2	46.9 ± 18.2	0.018	0.136

Data are shown as n (%) or mean ± SD or median (interquartile range). LV: left ventricular. LVMI: LV mass index. PWTd: posterior wall thickness in diastole. RWT: relative wall thickness. EDV: end-diastolic volume. EDVI: EDV index. EF: ejection fraction. SV: stroke volume. SVI: SV index. GLS: global longitudinal strain. DT: deceleration time. E′: peak early diastolic tissue velocity. A’ mean: peak late diastolic tissue velocity. E/E′: peak early filling over early diastolic tissue velocities ratio. LA: left atrial. LAVI max, pre-A, min: LA volume index maximal, at the onset of A wave, minimal. ef: emptying fraction. RV: right ventricular. RA: right atrial. TAPSE: tricuspid annular plane systolic excursion. FAC: fractional area change. RAV: RA volume. TR: tricuspid regurgitation. The *p*-value refers to the comparison between low and high NT-proBNP groups. * The comparison is adjusted for age, gender, BMI and creatinine. ^†^ The comparison is additionally adjusted for systolic blood pressure, use of loop and thiazide diuretics and beta-blockers. ^$^ Patients with chronic AF (n = 12) were excluded from this analysis.

**Table 3 biomedicines-11-00867-t003:** Association between log-transformed NTproBNP and echocardiographic measures in the overall HFpEF population.

	Model 1 * β (95% CI)	*p*-Value	Model 2 ^†^ β (95% CI)	*p*-Value
LVMI (g/m^2^)	7.63 (1.23; 14.02)	0.020	7.70 (0.57; 14.82)	0.035
PWTd (mm)	0.03 (−0.49; 0.56)	0.895	−0.11 (−0.69; 0.47)	0.699
RWT	−0.01 (−0.04; 0.01)	0.395	−0.02 (−0.05; 0.01)	0.139
LVEDV (mL)	−1.10 (−8.36; 6.17)	0.766	−0.31 (−8.45; 7.82)	0.939
LVEDVI (mL/m^2^)	−0.12 (−4.01; 3.76)	0.949	0.69 (−3.67; 5.04)	0.755
EF (%)	−2.00 (−4.12; 0.12)	0.065	−1.64 (−4.00; 0.71)	0.170
GLS (%)	−1.25 (−2.42; −0.08)	0.037	−0.63 (−1.97; 0.71)	0.355
DT (ms)	−15.6 (−31.37; 0.23)	0.053	−17.1 (−34.83; 0.53)	0.057
Lateral E′ (cm/s)	−0.13 (−0.91; 0.65)	0.747	−0.48 (−1.35; 0.39)	0.274
Septal E′ (cm/s)	−0.14 (−0.626; 0.33)	0.551	−0.10 (−0.63; 0.42)	0.697
Mean E′ (cm/s)	−0.23 (−0.79; 0.33)	0.415	−0.36 (−0.99; 0.27)	0.256
A′ mean (cm/s)	−2.04 (−2.81; −1.27)	<0.001	−1.94 (−2.81; −1.08)	<0.001
Lateral E/E′	3.54 (1.49; 5.59)	0.001	3.72 (1.45; 6.00)	0.002
Septal E/E′	3.08 (1.15; 5.01)	0.002	2.55 (0.44; 4.66)	0.018
Mean E/E′	3.11 (1.32; 4.90)	0.001	3.10 (1.11; 5.09)	0.002
Max LAVI (mL/m^2^)	8.82 (4.54; 13.09)	<0.001	9.51 (4.72; 14.30)	<0.001
Pre-A LAVI (mL/m^2^)	6.45 (2.86; 10.05)	0.001	7.04 (2.99; 11.1)	0.001
Min LAVI (mL/m^2^)	7.83 (4.78; 10.87)	<0.001	8.12 (4.79; 11.46)	<0.001
LA global ef (%)	−6.93 (−10.12; −3.74)	<0.001	−6.39 (−9.84; −2.93)	<0.001
LA passive ef (%)	−1.58 (−4.80; 1.64)	0.334	−0.38 (−4.01; 3.25)	0.837
LA active ef (%)	−5.66 (−9.47; −1.85)	0.004	−5.08 (−9.30; −0.87)	0.019
LA compliance	−0.89 (−1.31; −0.46)	<0.001	−0.87 (−1.34; −0.40)	<0.001
TAPSE (mm)	−1.04 (−2.36; 0.28)	0.121	−0.63 (−2.06; 0.79)	0.382
RV FAC (%)	−3.53 (−8.16; 1.10)	0.134	−3.32 (−8.64; 2.01)	0.219
RV strain	−2.22 (−6.28; 1.84)	0.275	−0.05 (−4.52; 4.42)	0.983
TR velocity (m/s)	0.03 (−0.19; 0.26)	0.769	−0.02 (−0.31; 0.27)	0.876
RAV max (mL)	11.70 (5.73; 17.67)	<0.001	9.76 (2.75; 16.77)	0.007

Data are shown as n (%) or mean ± SD or median (interquartile range). T2DM: Type 2 Diabetes Mellitus. LV: left ventricular. LVMI: LV mass index. PWTd: posterior wall thickness in diastole. RWT: relative wall thickness. EDV: end-diastolic volume. EDVI: EDV index. EF: ejection fraction. SV: stroke volume. SVI: SV index. GLS: global longitudinal strain. DT: deceleration time. E′: peak early diastolic tissue velocity. A′ mean: peak late diastolic tissue velocity. E/E′: peak early filling over early diastolic tissue velocities ratio. LA: left atrial. LAVI max, pre-A, min: LAv index maximal, at the onset of A wave, minimal. ef: emptying fraction. RV: right ventricular. RA: right atrial. TAPSE: tricuspid annular plane systolic excursion. FAC: fractional area change. RAV: RA volume. TR: tricuspid regurgitation. * The analysis is adjusted for age, gender, BMI and creatinine. ^†^ The analysis is additionally adjusted for systolic blood pressure, use of loop and thiazide diuretics and beta-blockers.

**Table 4 biomedicines-11-00867-t004:** Association between log-transformed NTproBNP and log-transformed CMR measures in 71 HFpEF patients.

	Median (IQR Range)	Model 1 * β (95% CI)	*p*-Value	Model 2 ^†^ β (95% CI)	*p*-Value
Myocardial T1 pre-contrast, ms	1005 (980–1046)	−0.21 (−3.68; 3.62)	0.908	−1.00 (−4.18; 2.17)	0.530
Myocardial T1 post-contrast, ms	354 (328–385)	−0.12 (−0.02; 0.26)	0.913	0.004 (−2.03; 2.04)	0.997
Extracellular volume fraction, %	27.6 (25.0–31.4)	1.74 (0.10; 3.48)	0.049	1.82 (0.19; 3.44)	0.029

* The analysis is adjusted for age, gender, BMI and creatinine. ^†^ The analysis is additionally adjusted for systolic blood pressure, use of loop and thiazide diuretics and beta-blockers.

## Data Availability

Not applicable.

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
