# Peer review of "Natriuretic Peptide Levels and Stages of Left Ventricular Dysfunction in Heart Failure with Preserved Ejection Fraction"

_biomedicines, 2023, doi:10.3390/biomedicines11030867_

Round 1

Reviewer 1 Report

This is an excellent work.

There is one issue: the hemoglobin level (table 1) is low (8.3 v. 8.0). No unit is given. Could this explain some of the other findings or is this completely unrelated? Could the iron status (serum iron, TSAT, ferritin) be relevant here? Although iron deficiency does not affect mortality, it plays an important role in HFpEF and functional outcome

Author Response

We would like to thank the reviewer very much for the kind words.

Regarding the issue mentioned by the reviewer:

Thank you very much for pointing this out. We have added the unit for haemoglobin (in mmol/l) in Table 1. As the majority of HFpEF patients were female, the mean haemoglobin values are within normal range.

Reviewer 2 Report

The authors investigated cardiac remodeling/dysfunction and myocardial tissue characteristics assessed by echocardiography and cardiac magnetic resonance in HFpEF patients in relation to natriuretic peptides levels. They concluded that higher NT-proBNP levels can identify HFpEF patients with worse LV stiffness because of more severe myocardial extracellular matrix remodeling, representing an advanced stage of HFpEF.

General comments
This is a manuscript addressing an important topic “Natriuretic Peptides Levels and Stages of Left Ventricular Dysfunction in Heart Failure with Preserved Ejection Fraction”. However, the discussion and conclusions drawn are only partly supported by the results. Some concerns need to be addressed.  

Specific comments

1)      Line 396: “Future studies may consider including HFpEF patients with low NT-proBNP values in whom high diastolic LV stiffness may be predominantly affected by increased cardiomyocyte stiffness instead of myocardial fibrosis and may also benefit from drug therapy”. Because half of the patients had high ECV >30% in this cohort, only increased cardiomyocyte stiffness would not explain the LV stiffness. The reviewer recommends a tone-downed version of the conclusions.

2)      Table 1: the unit is missing in “Hemoglobin”.

3)      Line 107: In the figure 1, the NT-proBNP was indicated as raw data, but not log-transformed one.

4)      Figure 2 would be redundant because all data are in the table 2.

5)      Figure 3: the correlation analysis would be also added to this fig. and methods.

Author Response

We would like to thank the reviewer very much for the evaluation of our manuscript and the thoughtful comments raised, which we will address point by point below.

Specific comments

1)      Line 396: “Future studies may consider including HFpEF patients with low NT-proBNP values in whom high diastolic LV stiffness may be predominantly affected by increased cardiomyocyte stiffness instead of myocardial fibrosis and may also benefit from drug therapy”. Because half of the patients had high ECV >30% in this cohort, only increased cardiomyocyte stiffness would not explain the LV stiffness. The reviewer recommends a tone-downed version of the conclusions.

Thank you for this constructive remark. We have deleted this sentence and rephrased it to:

"As NT-proBNP reflects the severity of cardiac dysfunction and remodeling in HFpEF, it may potentially serve a contributory role in improving pathophysiologic and therapeutic stratification in patients with HFpEF". Line 402 in the revised manuscript.

2)      Table 1: the unit is missing in “Hemoglobin”.

Thank you very much for pointing this out. We have added the unit (mmol/l) for hemoglobin to Table 1. 

3)      Line 107: In the figure 1, the NT-proBNP was indicated as raw data, but not log-transformed one.

Thank you for pointing this out. We have changed the text in Line 107 accordingly and deleted the word log-transformed: 

"The distribution of HFpEF patients in accordance to plasma NT-proBNP is shown in Figure 1". 

4)      Figure 2 would be redundant because all data are in the table 2.

We acknowledge the comment of the reviewer and agree. All data are also in Table 2. Therefore, we will delete Figure 2. 

5)      Figure 3: the correlation analysis would be also added to this fig. and methods.

Thank you very much for pointing this out. We have added the numerical value of the correlation analysis to the Figure en added the following sentence to the text (Line 176-178) in the results section:

"The linear relationship between log-transformed NT-proBNP and extracellular volume (ECV) is represented in Figure 2. Higher NT-proBNP plasma levels were significantly and independently associated with higher proportion of ECV, β = 1.74 (95% CI:0.10-3.48)".

This is also described in the method section (Line 370-371): 

"We also assessed associations of log-transformed NT-proBNP as continuous variable with echocardiographic measures and with log-transformed CMR-T1 mapping indices".